# 4-Ethylacetophenone from Potato Plants Repels *Phthorimaea operculella* and Inhibits Oviposition: A Sustainable Management Strategy

**DOI:** 10.3390/insects16040403

**Published:** 2025-04-11

**Authors:** Xinyu Ma, Junjie Yan, Guangyuan Su, Fathiya M. Khamis, Athanase Hategekiman, Yulin Gao

**Affiliations:** 1State Key Laboratory for Biology of Plant Diseases and Insect Pests, Institute of Plant Protection, Chinese Academy of Agricultural Sciences, Beijing 100193, China; sdaumaxinyu@163.com (X.M.); yanjunjie01@caas.cn (J.Y.); 17812723081@163.com (G.S.); 2International Center of Insect Physiology and Ecology (ICIPE), Nairobi P.O. Box 30772-00100, Kenya; fkhamis@icipe.org; 3Department of Zoology and Entomology, University of Pretoria, Hatfield 0028, South Africa; 4Rwanda Agriculture and Animal Resources Development Board RAB, Kigali P.O. Box 5016, Rwanda; athanase.hategekimana@rab.gov.rw

**Keywords:** potato tuber moth, repellents, plant volatile, electrophysiological behavioral responses

## Abstract

The potato tuber moth (*Phthorimaea operculella*) is a destructive pest that infests potato crops by boring into and eating the tubers. Although chemical pesticides are frequently employed to manage these moths, their efficacy is inconsistent and they can cause environmental damage. In this study, we investigated a natural alternative by examining specific volatiles emitted by potato plants that may repel the moths and inhibit oviposition. We identified five plant-based compounds, one of which, 4-ethylacetophenone, exhibited strong repellent effects against the moths at all tested concentrations and significantly reduced egg-laying behavior. This finding suggests that 4-ethylacetophenone could serve as an eco-friendly way to protect potato crops from *P. operculella*, providing a safer alternative to conventional pest control strategies.

## 1. Introduction

The potato tuber moth, *Phthorimaea operculella* (Zeller), is a worldwide oligophagous pest insect with a wide distribution and high reproductive capacity. It is one of the main pests on solanaceous crops, including potato and tobacco, and seriously damages potato production annually [1,2]. Depending on the ecological cultivation area, 5–13 generations occur annually in China, with overlapping generations. The potato tuber moth is a significant pest that can cause severe damage to potato crops, especially in polyculture areas where potatoes are grown in conjunction with tobacco. In regions like Yunnan Province in China, where polyculture practices are prevalent, the potato tuber moth causes particularly devastating impacts [3]. The adults of *P. operculella* frequently deposit their eggs on the stems and undersides of host leaves, as well as in soil cracks and on exposed tubers [4]. There are four instars, and the younger larvae (one to three instars) mine the leaves or tubers of host plants, often confined to a single tunnel [5,6]. The mature larvae overwinter, producing cocoons and turning into pupa in soil cracks below the host plant. Due to the changing climate and cropping pattern in China, *P. operculella* has strong adaptability, rendering effective monitoring, prediction, and management of this pest a crucial task in field control work [3,7].

Over the last two decades, chemical pesticides have been widely applied for pest management during outbreak periods [8]. Although the management methods for the potato tuber moth using resistant variety, sex pheromones and biological control via fungi have been studied in China [9,10,11]. Current research focuses on various aspects of the pest, including its genetics, development, and gut function [12,13]. Host plant volatiles have also been extensively studied as olfactory signaling molecules in insects. Herbivory induced plant volatiles (HIPVs) play a key role in the ecosystem by attracting natural enemies through air or inducing plant defense [14,15,16]. For example, *ß*-caryophyllene has been extensively implicated in deterring herbivores directly or attracting natural enemies of herbivores, including *Cotesia marginiventris*, *Aphidius gifuensis*, and *Trichogramma chilonis* [17,18,19]. HIPVs may also attract or repel the pest directly. Previous studies have characterized volatile compounds from potato and tobacco plants and evaluated their electrophysiological effects on *P. operculella*. These investigations aimed to identify potential chemicals for managing the potato tuber moth. Compounds such as heptanal, eucalyptol, nerol, 2-phenylethanol, and linalool have been extensively studied for their effects on the pest [13,20,21]. High-temperature (HT) stress and insect herbivory alter the volatile organic compound (VOC) profiles of plants. High-temperature pre-stressed plants emitted VOCs that reduced attractiveness to *P. operculella* moths but attracted more parasitoids, indicating altered plant–insect interactions that could be harnessed for pest management. *ß*-caryophyllene was repellent to *P. operculella*, while *Z*−3-hexen-1-ol acetate attracted both *P. operculella* and *T. chilonis* [22].

Our previous studies on the ovipositing effects of various potato varieties on *P. operculella* revealed that the Dahailao potato variety is preferred to the Pb08 potato variety by *P. operculella* for egg-laying. However, the specific volatile chemicals responsible for this preference remain unclear. To identify potential chemicals for managing *P. operculella*, we used GC-MS to isolate and ascertain the identity of five potato plant volatiles. Based on this detailed and comprehensive analysis, our current study aimed to compare the volatiles emitted by two potato varieties and screen five compounds with relatively high doses for electroantennogram (EAG) responses. Furthermore, we aimed to investigate how these active compounds regulate moth behavior.

## 2. Materials and Methods

### 2.1. Insects

Larvae of *P. operculella* were collected from the leaves of potato plants in the field in 2014, and then the *P. operculella* population was established at the Vegetable Pest Laboratory of the Institute of Plant Protection, Chinese Academy of Agricultural Sciences. They were kept in topless plastic containers (diameter × height = 8 × 10 cm), with the openings covered by gauze. Whatman no. 1 filter paper, manufactured by Hangzhou Special Paper Industry Co., Ltd., Hangzhou, China, was placed above the gauze, which was replaced daily to collect the eggs of *P. operculella*. Moths were reared at (26 ± 1) °C, (60 ± 10)% relative humidity (RH), and a 12L–12D photoperiod in a biological oxygen demand incubator (MLR-351H, Sanyo Electric Co., Ltd., Moriguchi, Osaka, Japan).

Pupae of *P. operculella* were collected and kept individually until emergence. One-day-old and two- to three-day-old *P. operculella* were selected for experiments.

### 2.2. Chemicals

Previous research by our group revealed that *P. operculella* exhibited significantly higher oviposition activity on the Dahailao potato variety compared to PB08. Notably, 4-ethylacetophenone was exclusively detected in Dahailao. All VOCs (≥95% purity) used in the experiments were commercially sourced (Table 1).

### 2.3. Antennal Responses of Phthorimaea operculella to the VOCs

The responses of mated *P. operculella* females to the five compounds were evaluated by the electroantennogram (EAG) method. VOCs (10 μL) were applied to a filter paper strip (4.0 × 0.5 cm), inserted into a glass Pasteur pipette, and sealed with parafilm for later use. One antenna of the mated female was excised at the base and 1 mm was removed from the tip with a sharp blade under the dissecting microscope. The prepared antenna was placed on two forks of a metal electrode coated with conductive glue. The pipette tip was inserted into a mixing tube with a small hole. A continuous stream of humidified air, delivered at a rate of 600 mL/min, was directed onto the prepared insect antenna. The current stimulation time was 0.1 s and the interval between continuous stimulation was not less than 1 min. The electrophysiological signals were amplified, output, collected by an IDAC-2 data acquisition controller, and analyzed by EAG Pro software (Syntech, Germany). Each antenna was stimulated in the following order: n-hexane, treatment (volatile organic compound), n-hexane. Each compound was applied to a single antennal preparation and tested from low to high dosages. The relative value of EAG was calculated using the following formula:EAG relative response value = E_m_ − CK_m_
where E_m_ (mV) is the average EAG reaction value of the tested compound and CK_m_ is the average EAG reaction value of the control (n-hexane). No fewer than five replicates were used for each compound.

### 2.4. Behavioral Response Assays

#### 2.4.1. Y-Tube Olfactometer Experiment

To determine whether the compounds that elicited significant electrophysiological responses in EAG could also induce behavioral responses in *P. operculella*, we conducted behavioral assays using Y-tube olfactometer experiments [23]. The setup included a glass Y-tube (internal diameter: 2.5 cm; base length: 22 cm; arm length: 10 cm; angle between arms: 70°) connected to odor source bottles containing either the test VOC (10 μL) or the control (n-hexane). Air was drawn into the bottles at 200 mL/min and female moths were introduced into the main arm. A choice was recorded if the moth crossed at least 5 cm into one of the side arms within 5 min and remained there for over 1 min. To avoid the position effect, the Y-tube directions were switched after 5 female moths were tested. The Y-tube was cleaned with water and absolute ethyl alcohol, then heated at 180 °C for 2 h. Experiments were conducted in darkness at 26 ± 2 °C, with at least 30 replicates per treatment.

#### 2.4.2. Cage Experiments

To complement the Y-tube olfactometer experiments, we conducted cage experiments to simulate field conditions. This approach allowed us to validate the short-term preferences observed in the Y-tube experiments and assess the compounds’ effects on behaviors. Before the experiments, the lures were soaked in ethyl alcohol for 24 h and then dried naturally at room temperature. After drying, the lures were stored in sealed bags. For the experiments, the lures were treated with 100 μL of the test compound or n-hexane (as a control) and then attached to sticky boards and hung in cages (40 × 40 × 40 cm). Twenty mated female moths were released into each cage, and the number of moths attracted to the lures was recorded after 16 h. Each compound was tested with at least five biological replicates. To evaluate the behavioral preferences of *P. operculella*, we calculated the selection rates for both the control (n-hexane) and the treatment (4-ethylacetophenone or 4-ethylacetophenone) in each experiment:$$\mathrm{Selection}\;\mathrm{rate}=\frac{X}{T+C}$$
where *T* is the number of insects attracted to treatment, *C* is the number of insects attracted to n-hexane, and *X* is the number of insects attracted to treatment or n-hexane.

### 2.5. Oviposition Choice Test

Experiments were conducted in plastic oviposition cans with a diameter of 10 cm and a height of 15 cm, following the method described by Mo et al. [24]. A hole (diameter = 1 cm) was made on the side of the can so that insects could be placed. Some holes were made at the bottom for air circulation. Plastic cans were sealed with gauze and tied with rubber bands. The filter paper was divided into four sections with a pencil in advance. A piece of filter paper, a stainless-steel net, and four lures were placed on top of the container [25,26]. When the test began, two lures containing 10 μL compound solution or n-hexane were placed on two opposite areas.

Two-day-old adult males and females were selected and placed in cages for mating in darkness. One hour later, the mating pairs were gently captured using a 5 mL centrifuge tube. Once mating was concluded, four pregnant female were carefully transferred to the oviposition jar, while the male was removed. After 16 h, the quantity of eggs in each area was counted and the cumulative number of eggs enumerated in the two opposing areas was calculated. No fewer than seven replicates were used for each dosage.

The percentage of eggs laid was calculated as follows:$$\mathrm{Percentage}\;\mathrm{of}\;\mathrm{eggs}\;\mathrm{laid}=\frac{X}{T+C}$$
where *T* is the number of eggs laid on the filter paper areas with the compound solution and *C* is the number of eggs laid with n-hexane, and *X* is the number of eggs laid to treatment or n-hexane.

### 2.6. Statistical Analysis

One-way analyses of variance (ANOVAs) were conducted to analyze the EAG response data for various dosages of each compound with the same mating status, followed by Duncan’s multiple range test (*p* < 0.05). Chi-squared tests were used to analyze the significance of differences in the Y-tube olfactometer experiment. The Wilcoxon signed-rank test was used to compare the selection rates between the control and treatment groups in the cage experiment. For the oviposition choice test, the Wilcoxon matched-pair signed-rank test was utilized to analyze the oviposition preferences of mated females. Statistical analyses of all data were performed using GraphPad Prism 9 software (GraphPad Software, San Diego, CA, USA).

## 3. Results

### 3.1. EAG Activity

The electroantennogram (EAG) responses were used to determine the active volatile components *P. operculella* can perceive and to measure the relationship between EAG responses and compound dosages. The mated female moths had the highest antennal responses to 3-ethylacetophenone, followed by 4-ethylacetophenone, geranylacetone, ethylbenzene, and 4-hydroxy-4-methyl-2-pentanone (Figure 1, Appendix A).

There was no significant difference in the EAG responses of mated females to 4-hydroxy-4-methyl-2-pentanone (*p* < 0.05) (Figure 1A) or ethylbenzene (Figure 1B). For geranylacetone, the EAG response of the antennae was significantly higher at test dosages of 10 and 100 μg than <1 μg (*p* < 0.05) (Figure 1C). Similarly, at test dosages of 10 and 100 μg, the electrophysiological responses were significantly elevated (*p* < 0.05) for *P. operculella* tested with 4-ethylacetophenone (Figure 1D) and 3-ethylacetophenone (Figure 1E) compared to other dosages.

Within the dosage range of 0.01 to 100 μg, the relative responses of female moths to 3-ethylacetophenone exhibited an overall increasing trend. For 4-ethylbenzylacetone, a distinct peak was observed, indicating that its value increased with dosage before subsequently decreasing.

### 3.2. Behavioral Responses of P. operculella Adults to the VOCs

The directional behavior of mated female moths was assessed using a Y-tube olfactometer (Figure 2). At 0.01 to 100 μg, all dosages of 4-ethylacetophenone exhibited a repellent effect. Notably, at 10 μg (*χ*^2^ = 8.758, df = 1, *p* = 0.003) and 100 μg (*χ*^2^ = 10.8, df = 1, *p* = 0.001), 4-ethylacetophenone demonstrated highly significant negative taxis, with selection rates of 24.2% and 20%, respectively (Figure 2A). Within the 0.01–100 μg dosage range, 3-ethylacetophenone demonstrated an attractive effect on female moths. At 1 μg, the selection rate was significantly highest (*χ*^2^ = 5.121, df = 1, *p* = 0.024), reaching 69.7%. At other doses, the female moths showed no significant behavioral preference (Figure 2B).

The 4-ethylacetophenone exhibited significant repellent effects at all tested dosages (*p* < 0.05). The selection rates for the control (n-hexane) were 57% at 1 μg, 76% at 10 μg, and 76% at 100 μg, while the rates for 4-ethylacetophenone were 43%, 24%, and 24%, respectively (Figure 2C). These findings indicate that 4-ethylacetophenone effectively repels *P. operculella*, with the repellent effect becoming more pronounced at higher dosages.

At all tested dosages (1 μg, 10 μg, and 100 μg), 3-ethylacetophenone did not significantly repel or attract *P. operculella* (*p* > 0.05) (Figure 2D). The selection rates for 3-ethylacetophenone were 47% at 1 μg, 55% at 10 μg, and 41% at 100 μg, respectively. These results suggest that 3-ethylacetophenone has limited influence on the behavior of *P. operculella*.

### 3.3. Oviposition Selection of P. operculella to the VOCs

We investigated the attraction activity of 4-ethylacetophenone in the oviposition of *P. operculella* females (Figure 3A). The number of eggs in the area of n-hexane was significantly higher than 4-ethylacetophenone at dosages of 1, 10, and 100 μg (*Z* = −2.293, *p* = 0.022; *Z* = −2.803, *p* = 0.005; *Z* = −2.366, *p* = 0.018). There was a significant difference among the number of eggs produced by different 4-ethylacetophenone doses (*p* < 0.05) (Figure 3B). The number of eggs produced decreased was fewest when the dose was 100 μg (36.86 ± 11.23).

## 4. Discussion

As *P. operculella* is one of the most challenging pests to manage in potatoes, growers have commonly relied on widespread and excessive use of insecticides [27,28]. Consequently, this practice has contributed to a rising incidence of insecticide resistance, particularly against organophosphates and carbamates, thereby necessitating an increasing demand for alternative, non-insecticidal solutions in managing potato tuber moth [29]. One potential solution entails the utilization of plant volatiles. From a biochemical point of view, plant volatiles play a key role in the behavior regulation of insects. These volatile organic compounds (VOCs) can either repel herbivores and pathogens or attract natural predators of herbivores, thereby functioning as a natural pest population management strategy in potato production [30].

We identified 4-ethylacetophenone as an important plant volatile that repels *P. operculella* and inhibits its oviposition behavior, selected from five potential plant volatiles. We explored the behavioral responses of *P. operculella* to selected VOCs to identify the most active compound and its effect on the pest, finding a repellent effect for 4-ethylacetophenone and an inhibitory effect on egg-laying, which provides a new perspective for understanding plant–herbivorous insect interactions. Our study identified 4-ethylacetophenone as the most active VOC among the tested compounds. This is a common volatile compound found in various plants, including hollyhock, ripening peaches, sorghum panicles, and carob tree, albeit with differing relative contents and proportions [31,32]. Our study found that 4-ethylacetophenone may play a significant role as an olfactory cue in modulating the behavior of *P. operculella*, as it showed significant repellent activity against the pest. This has also been observed in previous studies on the behavioral responses of *Spodoptera frugiperda* to the volatiles of *Ricinus communis*, where 4-ethylacetophenone was observed to be repellent to female *S. frugiperda* [33]. This compound with *E*-2-nonenal elicited a positive behavioral response from the three principal grain beetle pests in the UK (*Oryzaephilus surinamensis*, *Sitophilus granarius*, and *Cryptolestes ferrugineus*) [32]. Li et al. (2022) also demonstrated that 4-ethylacetophenone could significantly attract male parasitoid wasps (*Microplitis mediator*) at 1 and 10 μg/μL [34]. *Microplitis mediator* is a broad-spectrum larval endoparasitic wasp that widely attacks key lepidopteran pests of agricultural importance, such as *Helicoverpa armigera* and *Mythimna separata* [35]. Research on utilizing *Galerucella placida* Baly (Coleoptera: Chrysomelidae) for controlling the Polygonaceae weeds *Rumex dentatus* and *Polygonum glabrum* has shown that female *G. placida* prefer insect-damaged plants of both weeds over other treatments. Further studies have uncovered that the insect-damaged weeds emitted increased levels of 4-ethylacetophenone. This compound, along with other key components, has been confirmed to form two blends of volatile compounds that can attract the biocontrol agent during the early vegetative stage of these two weeds [36]. Here, we propose a hypothesis that 4-ethylacetophenone is a compound released by plants when they are attacked by pests, possessing dual functions: both repelling pests and attracting natural enemies.

Electroantennogram findings suggest that both 3-ethylacetophenone and 4-ethylacetophenone may play significant roles as olfactory cues in modulating the behavior of the potato tuber moth. Although the electrophysiological responses of female antennae to 3-ethylacetophenone and 4-ethylacetophenone were higher than those of other compounds, 3-ethylacetophenone did not trigger stabilized behavioral responses in adult *P. operculella*. This may be due to the structural similarity of the two compounds, which can induce an EAG response, but the insect’s behavioral response requires additional physiological reactions apart from those of the antennae [37].

In future studies, we can delve deeper into the response of odor receptors to specific compounds. For instance, SlituOR19, an odorant receptor identified in *S. litura*, responds slightly to 4-ethylacetophenone [38]. The odorant receptor ApisOr23 from *Acyrthosiphon pisum* also responds to 4-ethylacetophenone with a relative response intensity of 44.18 ± 5.03 nA [39]. The ability of 4-ethylacetophenone to activate odorant receptors in both *S. litura* and *A. pisum* demonstrates its cross-species relevance in insect olfaction. However, the relatively weak responses of these receptors to 4-ethylacetophenone suggest that it may not be the main ligand for these receptors. Despite this, the potential of 4-ethylacetophenone to influence insect behavior makes it a promising candidate for pest management. Future research should focus on elucidating its mechanisms of action, particularly in *P. operculella*, to develop more effective and sustainable pest control strategies.

In the natural environment, insects are exposed to a complex blend of volatile olfactory cues that are often composed of multiple chemicals. The results of individual chemical studies may not fully reflect the behavior of insects in real ecological environments. However, certain chemicals may be more important than others in recognition and orientation processes [40]. Thus, a single chemical may have a limited repellent effect, but it can still play a key role in the process. Future studies can focus on strengthening the repellent effect of chemical mixtures on the behavior of potato tuber moths. In the practical application of pest control, the synergistic effect of various chemical substances should be considered comprehensively.

The results of our study provide new insights into potential strategies for sustainable and ecologically friendly management of the potato tuber moth. Compared with traditional insect repellents, 4-ethylacetophenone has significant advantages. It directly repels females and is environmentally friendly. It has great potential as a green product for controlling potato tuber moths. It is also possible to combine 4-ethylphenylethenone with known plant volatiles or sex attractants to achieve more effective control [41,42,43]. For example, azadirachtin and eucalyptol together reduced oviposition by 56.3% [20]. We propose investigating the push–pull effects of 4-ethylacetophenone with these compounds in field conditions, as well as exploring its synergistic effects with other repellents. This approach could lead to more sustainable and efficient pest management strategies.

Insects navigate the intricate process of host identification and location by interpreting complex blends of plant volatiles, which are composed of a diverse array of compounds in varying proportions [44]. While the presence of a single compound can indeed influence an insect’s choice of location, the synergistic effect of multiple compounds within these blends tends to exert a more consistent and reliable influence on the behavioral regulation of *P. operculella* [45]. For future research, it is important to investigate the synergistic impacts of diverse, active compounds when combined in varying concentration ratios. Such research should assess how these compound blends regulate the behavioral responses of the potato tuber moth both under controlled laboratory circumstances and in the more intricate field scenario. Moreover, beyond olfaction, the visual, tactile, and gustatory senses also play a crucial role in modulating the behavioral responses of *P. operculella* [46,47,48]. For example, through the comparison of integrated control technologies, a study found that the combination of insecticidal lamps and sex pheromones provided the best control effect against the potato tuber moth [49]. By integrating olfactory cues with visual and tactile stimuli, a more comprehensive understanding of the tri-trophic interactions among *P. operculella*, its natural enemies, and its host plant can be achieved. This integrated methodology will be conducive to developing powerful attractants and repellents customized to the sensory preferences of potato tuber moth, thereby augmenting the efficiency of pest management strategies.

The implementation of innovative pest management strategies for control of the potato tuber moth has become essential for sustaining agricultural practices in China. Therefore, the integration of host plant volatiles, such as 4-ethylphenylethenone, with additional attractants like cis-jasmone and sex pheromone to establish a push–pull management strategy may offer a more sustainable, non-chemical alternative for pest control. This discovery provides growers with a vital option for managing potato pests, particularly in light of the declining effectiveness of conventional insecticide programs due to rising insecticide resistance.

## Figures and Tables

**Figure 1 insects-16-00403-f001:**
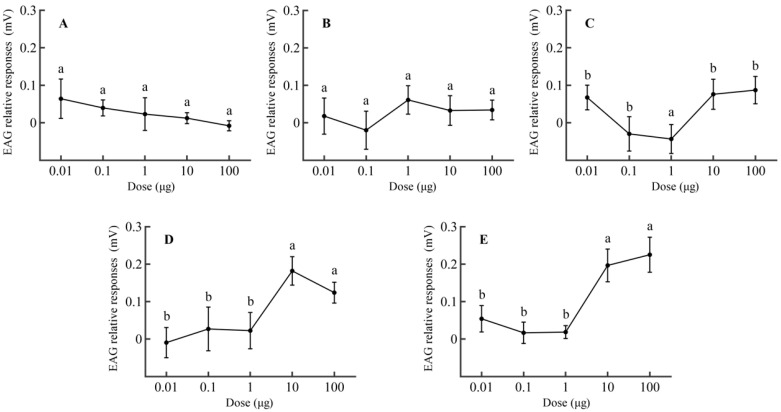
EAG responses (±SEM) of mated females of *P. operculella* to compounds at different dosages (*n* > 5). (**A**) 4-Hydroxy-4-methyl-2-pentanone, (**B**) ethylbenzene, (**C**) geranylacetone, (**D**) 4-ethylacetophenone, (**E**) 3-ethylacetophenone. Different letters indicate significant differences between the dosages for the same compounds (one-way ANOVA, Duncan’s multiple range test, *p* < 0.05).

**Figure 2 insects-16-00403-f002:**
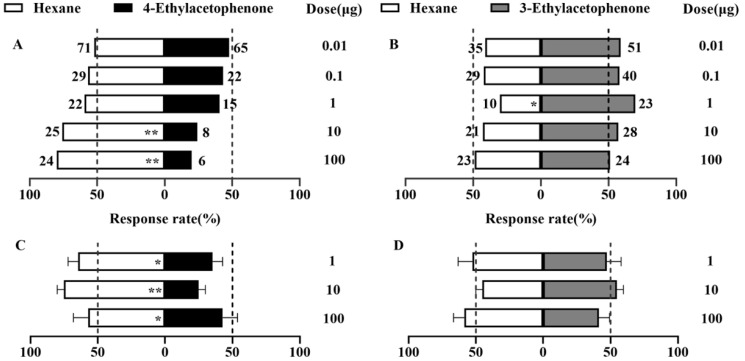
Behavioral responses of mated *P. operculella* females to the two compounds at different dosages in the Y-tube olfactometer experiment and cage experiment. (**A**,**B**) Y-tube olfactometer experiment, (**C**,**D**) cage experiment. (**A**) 4-ethylacetophenone, (**B**) 3-ethylacetophenone, (**C**) 4-ethylacetophenone, (**D**) 3-ethylacetophenone. The chi-squared test was used to analyze the significance of differences in the Y-tube olfactometer experiment (*n* > 30) and Wilcoxon signed-rank test to compare the selection rates in the cage experiment (*n* > 5). * and ** indicate significant differences (*p* < 0.05) and extremely significant differences (*p* < 0.01), respectively.

**Figure 3 insects-16-00403-f003:**
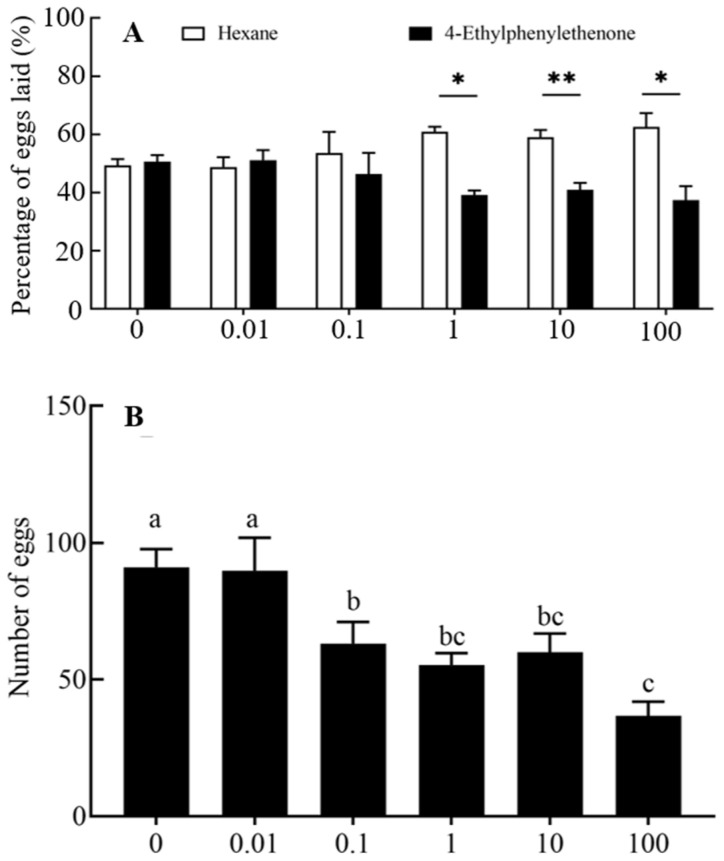
Preference for oviposition and the egg production laid by mated female *P. operculella* in response to varying dosages of 4-ethylacetophenone (*n* > 7). (**A**) Oviposition preference of mated females of *P. operculella* to 4-ethylacetophenone at different doses. The numbers (%) indicate the mean of eggs laid on the areas laced with the compound or n-hexane. Data were analyzed using the Wilcoxon matched-pair signed-rank test. * *p* < 0.05; ** *p* < 0.01. (**B**) Egg production of *P. operculella* in response to 4-ethylacetophenone at different doses. Different letters indicate significant differences between the dosages for the same compounds (one-way ANOVA, Duncan’s multiple range test, *p* < 0.05).

**Table 1 insects-16-00403-t001:** Standardized information on substances.

Standard Compounds	CAS No.	Purity (%)	Manufacturer
3-Ethylacetophenone	22699-70-3	98	Yuanye Bio-Technology Co., Ltd.
Geranylacetone	3796-70-1	98	
Ethylbenzene	100-41-4	95	Macklin Biochemical Co., Ltd.
4-Ethylacetophenone	937-30-4	96	Acmec Biochemical Technology Co., Ltd.
4-Hydroxy-4-methyl-2-pentanone	123-42-2	99	
n-Hexane	110-54-3	95	Aladdin Biochemical Technology Co., Ltd.

Each compound was dissolved and diluted to obtain concentrations of 10^−3^, 10^−2^, 10^−1^, 10^0^, and 10^1^ μg/μL with n-hexane and stored at −80 °C.

## Data Availability

The data supporting the conclusions of this article are provided within the article. The electrophysiological response data of *P. operculella* were listed in Appendix A.

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
