# Peer review of "4-Ethylacetophenone from Potato Plants Repels *Phthorimaea operculella* and Inhibits Oviposition: A Sustainable Management Strategy"

_insects, 2025, doi:10.3390/insects16040403_

Round 1
Reviewer 1 Report
Comments and Suggestions for Authors
I think this ms is good enough to be published without any revisions.
Author Response
Dear Reviewer,
Thank you very much for your positive feedback and kind assessment of our manuscript. We are truly delighted and encouraged to learn that you consider our work to be in a publishable state without further revisions. Your recognition is a great motivation for us and strengthens our confidence in continuing our research efforts.
We will proceed with the publication process as per the journal's requirements. Once again, we deeply appreciate your time and valuable insights.
Sincerely,
MaXinyu.
Reviewer 2 Report
Comments and Suggestions for Authors
The manuscript “Repelling potato tuber moth, Phthorimaea operculella Zeller (Lepidoptera: Gelechiidae) using a plant volatile: a potential management strategy “present serious writing problems (see attached pdf). In addition, title does not say exactly what authors made. Novelty of the study is somewhat questionable since previous studies have already study this topic. The methodological approach is not sufficiently strong, for example, is not necessary the chemical description, too long. Authors did not say why they tested the behavioral response of P. operculella only to toward specific VOCs, in Y-tube olfactometer. In result, in some parts statistical analylisis is confusing. Discussion. The hypothesis that the authors mention lack of scientific support. After line 279 discussion in too speculative. Finally, language paper must be carefully revised by a native English speaker. The manuscript should not be published in its current form. After a major revision manuscript could be submitted elsewhere.

Language paper must be carefully revised by a native English speaker.
Author Response
Response to the reviewers' comments
Comments 1:The manuscript “Repelling potato tuber moth, Phthorimaea operculella Zeller (Lepidoptera: Gelechiidae) using a plant volatile: a potential management strategy “present serious writing problems (see attached pdf).
Response 1: Thank you for your comments. I am so sorry about the writing problems, and we have made a thorough revision of the article according to your feedback. some Please see the new version). In addition, we have also corrected the formatting issues in the article (References)
- Line 2-4:
We have revised the title to: "4-Ethylacetophenone from potato plants repels Phthorimaea operculella and inhibits oviposition: a sustainable management strategy" , more details please see the Response 2.
- Line 23: "while 10 and 100 μg of 4-ethylacetophenone significantly repelled operculella."
We added some details revised the conclusions to make the results clearer(Line 22-25).
- Line 34: "being"
It was changed to "is". (Line 37)
- Line 35: "it’s"
It was changed to "it is". (Line 39)
- Line 37: "year.[1,2] "
We adjusted the position of the citations.(Line 42)
- Line 42: "common "
It was deleted, and we made changes to the language. (Line 46-47)
- Line 51: "pest"
It was deleted, and we made changes to the language. (Line 55-56)
- Line 53: "China10,11. Currently, research focuses on the pest’s genetics"
We removed the non-standard format, and we made changes to the language. (Line 59-60)
- Line 54: "others[10–13]"
We added"."(Line 60)
- Line 55: "herbiory"
It changed to " Herbivory "
- Line 56: "fense. [14–16].
"."was deleted. (Line 63)
- Line 60: "Prior".
It was changed to"Previous"
- Line 63: "on have"
We have revised this sentence(Line 70-71)
- Line 67: "management.."
"." was deleted. (Line 75)
- Line 93-103:
We have streamlined the descriptions of the methodological approach(Lines 100–103).
- Line 113: "knife"
We revised it to a " sharp blade "(Line 124)
- Line 120: "order::"
":" was deleted. (Line 131)
- Line 140: "5cm"
We added a blank.
- Line 150: "in the plastic"
" The " was deleted. (Line 188)
- Line 154: "Put a piece"
"Put" was changed to "Place"(Line 192)
- Line 159: "5mL"
We added a blank. (Line 197)
- Line 162: " enumerated."
" enumerated." was changed to "counted"(Line 200)
- Line 184: " 4-Hydroxy-4-methyl-2-pentanone (P<05)"
We changed the words" Hydroxy " " P "to lowercase. (Line 224-225)
- Line 215: "P. operculella females"
We italicized the word. (Line 271)
- Line 223: " P. operculella"
We italicized the word. (Line 278)
- Line 233: " resorted"
We replaced the word with" relied on "(Line 288)
- Line 242: " In this study, we successfully selected one plant volatile that can regulate the behavior of P. operculella from five potential plant volatiles."
We made the results clearer and highlighted the dual role of 4-ethylacetophenone(Line 297-300)
- Line 268-270: " Here we propose a hypothesis that 4-ethylacetophenone is a compound released by plants when they are attacked by pests, possessing dual functions: both repelling pests and attracting natural enemies"
We did not change this part because there is the literature showed that 4-ethylacetophenone can attract Microplitis mediator, which is the parasitic enemy of the larva of the Lepidoptera pest. We have proved the activity of 4-ethylacetophenone repelling Phthorimaea operculella and inhibiting oviposition. Besides, it is derived from the host plant. In future studies, we can design an experiment to show that insect pests increase its volatilities and that it attracts parasitic wasps, explaining its important role in the three trophic levels.
- Line 279-285:
We have revised the part you mentioned to make it more believable(Line 325-327)
- Line 299-300:
We give an example to support our mind. Besides, we have introduced literature in Line 365.
- Line 312-316: " resorted"
We give an example to support our minds and introduce more literature. Literature [46–48] can also give scientific support. (Line 381-383)
- Line 329-436 (References)
We have corrected the formatting issues in the article (References)
Comments 2: In addition, title does not say exactly what authors made.
Response 2:Thank you for your valuable comments. We have revised the title to: "4-Ethylacetophenone from potato plants repels Phthorimaea operculella and inhibits oviposition: a sustainable management strategy"This new title explicitly highlights the key findings of our study, including the repellent and oviposition-inhibiting effects of 4-ethylacetophenone.
Comments 3:Novelty of the study is somewhat questionable since previous studies have already study this topic.
Response 3: Thanks for your recommendation. While previous studies have explored plant volatiles for pest management, our work provides three novel contributions:
- First identification of 4-ethylacetophenone as a repellent specific to operculella (Figure 2A).
- Dual functionality: Unlike existing studies focusing solely on attraction/repellence, we demonstrate its oviposition suppression effect at high doses (Figure 3).
- Mechanistic hypothesis: We propose that 4-ethylacetophenone may act as an herbivore-induced plant volatile (HIPV) attracting natural enemies (Lines 315-323), a hypothesis not previously tested for this compound.
Comments 4: The methodological approach is not sufficiently strong, for example, is not necessary the chemical description, too long.(Line93-103)
Response 4: We sincerely thank you for this constructive feedback. Our long description of the method was to make it clear where the compound came from, but now we've changed it in light of your suggestions. we have streamlined the descriptions of the methodological approach(Lines 100–103). Besides, we also removed the descriptions of detailed chemical descriptions and unnecessary technical details in behavioral response assays (Lines 139–152).
Comments 5: Authors did not say why they tested the behavioral response of P. operculella only to toward specific VOCs, in Y-tube olfactometer.
Response 5: Thank you for your valuable comments. To complement the Y-tube olfactometer experiments and provide more evidence, we have added cage experiments to validate the behavioral responses of P. operculella to the selected VOCs. The cage experiments were conducted under controlled conditions, and the results consistently supported the findings from the Y-tube olfactometer tests. This addition enhances the reliability of our conclusions(Lines 170–187).
Comments 6: In result, in some parts statistical analylisis is confusing.
Response 6: Thank you for your careful review. We have added some details in the statistical analysis to make the results more clear(Line 182–187 ; Line 211–214).
Comments 7: Discussion.The hypothesis that the authors mention lack of scientific support. After line 279 discussion in too speculative.
Response 7: Thank you for your kindly comments. We have revised the part you mentioned and introduced more literatures to support our mind (Line 240-352 ; Lines 367-371; Lines 380-382)
Comments 8: Finally, language paper must be carefully revised by a native English speaker. The manuscript should not be published in its current form. After a major revision manuscript could be submitted elsewhere.
Response 8: Thanks for your recommendation. The paper have already revised by the native English speaker, and we made careful corrections to the grammatical problems this time. We hope that the revised version meets your expectations.
Reviewer 3 Report
Comments and Suggestions for Authors
Dear authors,
the study is interesting, it is very clear and structured, small adjustments could be considered (see details in the attached document)

Author Response
Response to the reviewers' comments
Comments 1:Line42: space before each cite (all document) please.
Response 1: Thanks for your careful checks. We have corrected them all.
Comments 2:Line53/56: check, pelase.
Response 2: Thanks for your kindly recommendation. We have checked and corrected this problem in the manuscript.
Comments 3:Line116/120: delete
Response 3: Thanks for your careful checks. We have deleted them all.
Comments 4:Line119/137: space
Response 4: Thanks for your recommendation. We added space here.
Comments 5:Line195:inset here (X) the tested product for each figure; the quality could be improved
Response 5: Thanks for your recommendation. In response to your comment about labeling the compound directly in the figure, I have taken your suggestion into serious consideration. However, due to the compound's lengthy name, adding it directly to the figure might result in overcrowding and compromise the overall clarity and visual appeal of the figure. The compound name has been clearly indicated in the figure caption. Therefore, I have refrained from making this change for now. Besides, we have adjusted the figure to make it clearer according to your kind suggestion.
Comments 6:Line211/216: is the control??? write, control –; change the scale interval, please to (0-25-50-75-100 ) in both figures; if hexane is a negative control, write it as a control throughout the text, otherwise it would look like hexane is one of the compounds tested. (text and figures)
Response 6: Thank you for your valuable comments. To highlight whether the compound is attractive or repellent, we set the scale interval like that. Besides, according to your valuable suggestion, we have emphasized n-hexane as a control throughout the article.
Comments 7: Line215/223 italic
Response 7: Thanks for your careful checks. We have corrected them all.
Comments 8: Line222 write here (x) the product name, please.
Response 8: Thanks for your kind recommendation. We did not set the name of compound because there are the names in the top of this figure. It would be a bit repetitive to illustrate it again in the figure.
Round 2
Reviewer 2 Report
Comments and Suggestions for Authors
no comments